

# Deep-learning jets with uncertainties and more

Sasha Bollweg[1], Manuel Haußmann[2], Gregor Kasieczka[1],
Michel Luchmann[3], Tilman Plehn[3*] and Jennifer Thompson[3]

**1** Institut für Experimentalphysik, Universität Hamburg, Germany
**2** Heidelberg Collaboratory for Image Processing, Universität Heidelberg, Germany
**3** Institut für Theoretische Physik, Universität Heidelberg, Germany

★ plehn@uni-heidelberg.de

## Abstract

Bayesian neural networks allow us to keep track of uncertainties, for example in top tagging, by learning a tagger output together with an error band. We illustrate the main features of Bayesian versions of established deep-learning taggers. We show how they capture statistical uncertainties from finite training samples, systematics related to the jet energy scale, and stability issues through pile-up. Altogether, Bayesian networks offer many new handles to understand and control deep learning at the LHC without introducing a visible prior effect and without compromising the network performance.


# 1 Introduction

Modern machine learning has recently gained significant impact in many directions of LHC physics. While boosted decision trees and relatively simple networks have been used in particle physics for more than 30 years, new technological developments suggest using deeper networks in a wide range of analysis tasks. The most active field in this direction has, for a long time, been subject physics and jet tagging [1]. Here, multi-variate analyses of high-level observables are currently being replaced by deep neural networks with low-level observables. This is a natural next step given our improved understanding of subjet physics both experimentally and theoretically, combined with the rapid development of standard machine learning tools. The inspiring aspect of this transition is whether deep learning will be merely used to slightly improve existing analysis techniques, or if we can apply it to do something new.

An early application of standard deep learning techniques to subjet physics uses image recognition tools on so-called jet images or maps in the rapidity vs azimuthal angle plane [2,3]. The most relevant information from the calorimeter then has to be combined with tracker output, challenging at least the simplest image recognition because of the different geometric resolution of the calorimeter and the tracker [4]. Benchmarks processes for jet images or alternative networks include quark-gluon discrimination [5,7–10], $W$-tagging [11,12], Higgs-tagging [13,14], or top-tagging [15–22]. By now this relatively straightforward classification task can be considered solved, at least at the level of tagging performance [23, 24]. The remaining open questions, which need to be studied before we can widely apply these kinds of taggers to standard LHC analyses, are related to systematics [25], general uncertainties [26], stability, weakly supervised learning [27–34], understanding the relevant physics input [35–39], and other LHC-specific issues which do not automatically have a counterpart in modern machine learning.

In particle physics applications, machine learning used for classification will not only be judged by the best performance on a standard data set, but by a mix of performance, properly defined uncertainties, and stability. While, for example, including error bars in classification tasks is not generally established [40,41], Bayesian neural networks (BNNs) do offer a new analysis opportunity in LHC applications: an event-by-event estimate of the uncertainty on the classification output. To illustrate the relevance of this information, if a network classifies an event as 60% signal or 40% background, the benefit of this piece of information rests on the uncertainty of that per-cent value. With an error bar of ±1% the 60% signal probability might well be useful for an analysis, an error bar of ±30% simply means that the given event is not going to be useful at all. Technically, Bayesian networks extract this information by not only providing a single output variable, but a distribution of the network output. With minimal assumptions, this distribution can then be translated into an uncertainty band on the network output.

In this paper we will, for the first time, use Bayesian networks for standard classification with uncertainties in the uncertainty-obsessed field of particle physics, namely top tagging [42]. In Sec. 2 we introduce Bayesian networks and discuss how they can be applied to a simple classification task. In Sec. 3 we relate some of their features to open questions in particle physics applications and relate Bayesian networks to the usual deterministic networks, where dropout and L2-regularization are essentially Bayesian features. For our application we also show how the output of Bayesian networks can be related to the frequentist approach of sampling many taggers. Finally, in Sec. 4 we test Bayesian versions of an image-based and a 4-vector-based top tagger in a more realistic setting. We study the ability of the network to track uncertainties due to the limited size of the training sample and due to systematics like the jet energy scale. An interesting aspect is that we can separate the leading systematic

uncertainty which is correlated with a shift of the mean network output, and a sub-leading uncorrelated systematic uncertainty. Finally, we show how the Bayesian network offers a new handle to test the stability of a classification network, for instance in the presence of pile-up.

## 2  Machine learning with uncertainties

Applying machine learning to classification tasks not only offers a way to extremely efficiently predict properties, for example of jets, it also allows us to define jet-by-jet uncertainty estimates on the tagging output. Possible sources of uncertainty in a particle physics framework include

- finite, but perfectly labeled training samples. In LHC analyses this corresponds to a statistical uncertainty in the classification output, for instance due to finite MC statistics;

- inconsistencies in the training data and their labels. In LHC analyses those correspond to systematic uncertainties on the classification or tagging output;

- differences between the training and test samples. In LHC analyses they arise from Monte Carlo simulation or control regions in data to the signal region and would again be treated as systematics at the classification output level.

The deep learning literature [43] defines two kinds of uncertainty: (i) epistemic or model uncertainty describes the lack of statistics and can therefore be expected to decrease with more data; (ii) aleatoric errors from noise in the data, which cannot be reduced by using more data. It makes sense to separate homoscedastic (universal) and heteroscendastic (input-dependent) noise. If we look at the scaling with increased data sets, this distinction corresponds exactly to statistical and systematic uncertainties in the LHC conventions.

It is crucial to notice that all three sources of uncertainties listed above are induced by either not perfect training data or by a not perfect match between training data and testing data. They are not uncertainties on the form of the actual network, which we simply consider a mathematical relation between network input and network output. Nevertheless, they all describe statistical or systematic uncertainties on the tagging output which have to be considered in the actual analysis.

### 2.1  Bayesian neural networks

Like all classifying neural networks, BNNs [44–49] relate training data $D$ to a known output or classifier $C$ through a set of network parameters $\omega$. Bayes' theorem then defines the (posterior) probability distribution for the parameters $p(\omega|\{D, C\})$ from the general relation

$$p(\omega|\{D, C\})\, p(\{D, C\}) = p(\{D, C\}|\omega)\, p(\omega). \tag{1}$$

In this form $\{D, C\}$ describes the combination of training inputs and network outputs for fully supervised learning. If we consider the training data $D$ as given, we can omit it in the shorter form

$$p(\omega|C) = \frac{p(C|\omega)\, p(\omega)}{p(C)}. \tag{2}$$

We can think of the prior $p(\omega)$ as the distribution of the model parameters before training on the data set $D$ and are free to choose it, for example, to be a Gaussian. The model evidence $p(C)$ serves as a normalization constant for the (posterior) probability distribution $p(\omega|C)$. The probability $p(\omega|C)$ allows us to predict the network output $c^*$ for a new test data point,

$$p(c^*|C) = \int d\omega\, p(c^*|\omega, C)\, p(\omega|C). \tag{3}$$

This line of argumentation immediately leads us to the main question behind this paper: can we define and determine a network output which is not just one number, like a signal probability, but a signal probability distribution in $\omega$ on a jet-by-jet level?

The technical problem with Eq.(3) is that we usually do not know the closed form of $p(\omega|C)$, even if it is implicitly encoded in our neural network. On the other hand, we can approximate it in the sense of a distribution and combine with a test function $p(c^*|\omega)$ [50],

$$\int d\omega \, p(c^*|\omega) \, p(\omega|C) \approx \int d\omega \, p(c^*|\omega) \, q(\omega). \tag{4}$$

The agreement between $p(\omega|C)$ and such an approximation $q(\omega)$ is given by the Kullback-Leibler divergence,

$$\text{KL}[q(\omega), p(\omega|C)] = \int d\omega \, q(\omega) \, \log \frac{q(\omega)}{p(\omega|C)}. \tag{5}$$

It vanishes if the two functions are identical almost everywhere and is positive otherwise. We can use Bayes' theorem to re-write it as

$$\begin{aligned}
\text{KL}[q(\omega), p(\omega|C)] &= \int d\omega \, q(\omega) \, \log \frac{q(\omega)p(C)}{p(C|\omega)p(\omega)} \\
&= \text{KL}[q(\omega), p(\omega)] + \log p(C) \int d\omega \, q(\omega) - \int d\omega \, q(\omega) \, \log p(C|\omega). \tag{6}
\end{aligned}$$

The second term only includes the normalization of $q(\omega)$ and is of no particular interest given that we have normalized $q(\omega)$ as a probability distribution. The third term is the usual expected likelihood, which we can use to work with in a frequentist sense, if we want to avoid the Bayesian prior altogether. In our framework we minimize the KL-divergence to construct a $q(\omega)$ approximating $p(\omega|C)$ in Eq.(3), so our loss function becomes

$$L = \text{KL}[q(\omega), p(\omega)] - \int d\omega \, q(\omega) \, \log p(C|\omega). \tag{7}$$

In pushing $L$ to its well-defined lower limit, the first term requires $q(\omega)$ to be close to an assumed, for instance Gaussian, prior $p(\omega)$. If we assume that $q(\omega)$ and $p(\omega)$ are both Gaussians described by their respective $\mu$ and $\sigma$ and we only consider a single weight, we can simplify the KL-divergence to

$$\text{KL}[q(\omega), p(\omega)] = \log \frac{\sigma_p}{\sigma_q} + \frac{\sigma_q^2 + (\mu_q - \mu_p)^2}{2\sigma_p^2} - \frac{1}{2}. \tag{8}$$

The last term in Eq.(7) needs to be minimized once we know the likelihood $p(C|\omega)$ for a given $C$ and evaluated as a function of $\omega$. Technically, this requires a variation of the parameters which give the functional form of the Gaussian $q_{\mu,\sigma}$. That means that we compute the derivative of $L$ with respect to $\mu$ and $\sigma$, leading for example to the condition

$$\frac{\partial}{\partial \mu} \int d\omega \, q_{\mu,\sigma}(\omega) \, \log p(C|\omega) = 0. \tag{9}$$

Once we determine the approximate probability distribution $q_{\mu,\sigma}(\omega)$ from Eq.(7), we can use it to solve Eq.(3) by Monte Carlo integration and find the predictive mean for the test sample,

$$p(c^*|C) \approx \int d\omega \, p(c^*|\omega) \, q_{\mu,\sigma}(\omega) \approx \frac{1}{N} \sum_j^N p(c^*|\omega_j(\mu,\sigma)) \equiv \mu_{\text{pred}}. \tag{10}$$

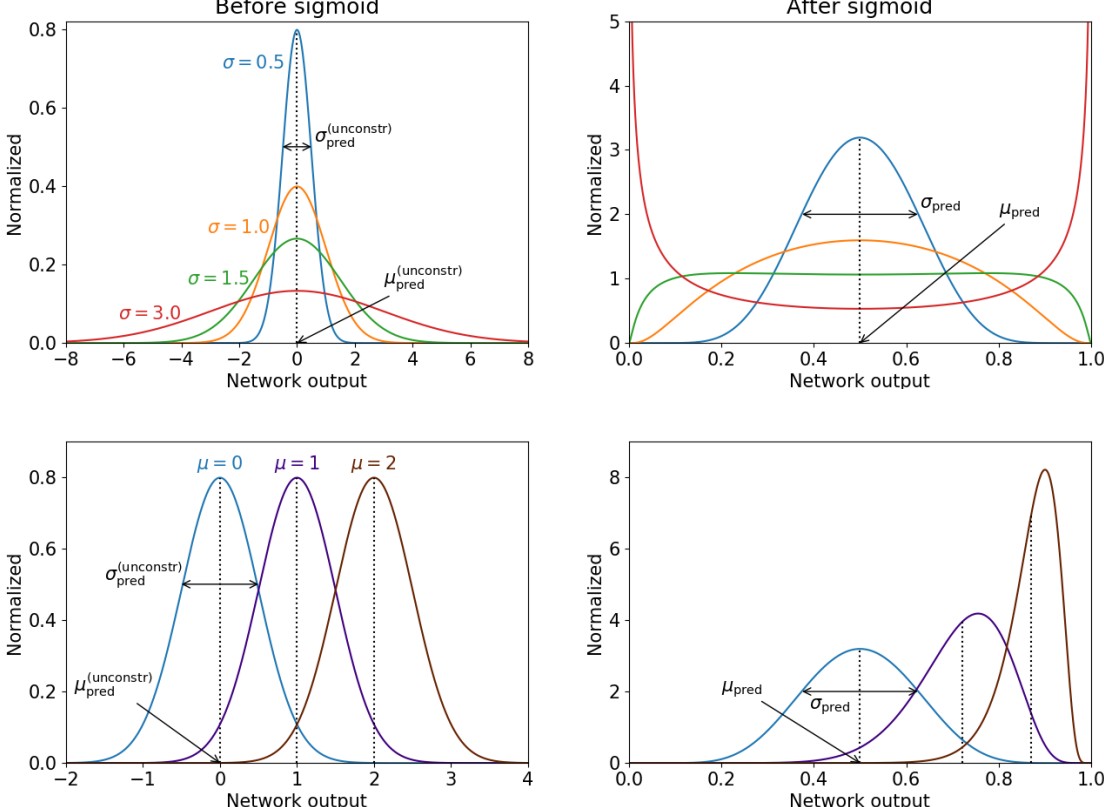

Figure 1: Effect of the sigmoid transformation on normal distributions with the same mean but different widths (upper) and on the same width but different means (lower).

To compute this mean we use $N$ sets of weights $\{\omega\}$, effectively corresponding to $N$ networks with different weights. Assuming a Gaussian probability distribution we also need the spread of the $N$ sets of weights, or the predictive standard deviation

$$\sigma^2_{\mathrm{pred}} = \frac{1}{N} \sum_j^N \left[ p(c^*|\omega_j(\mu,\sigma)) - \mu_{\mathrm{pred}} \right]^2. \qquad (11)$$

This way the BNN returns not only a central value $\mu_{\mathrm{pred}}$ for the classifying outcome, but also an jet-by-jet uncertainty estimate for this classification outcome $\sigma_{\mathrm{pred}}$.

## 2.2 Probabilities

We can numerically test this behavior with a toy BNN, analyzing jet images with $40 \times 40$ pixels. It does not include a convolutional layer and only consists of two fully connected hidden layers with a ReLu activation function, each with 100 units, and an output layer with one unit and a sigmoid activation function. It delivers a scalar output. For the BNN version of this toy network we use the TensorFlow Probability library [51] and its DenseFlipout Layer [52]. We have convinced ourselves that sampling 100 times from the weight distributions gives us stable results. Unless specified otherwise, we train our toy model on 100k top and 100k QCD jets. The BNN property means that we are not only interested in the values of the network output, but in the output distributions, as we will discuss in Sec. 3.2. Our toy BNN is trained to distinguish a public set of 600k top jet and QCD jet images each [17], which were generated with PYTHIA8 [53] for an LHC energy of 14 TeV and without pile-up or multiple interactions. As

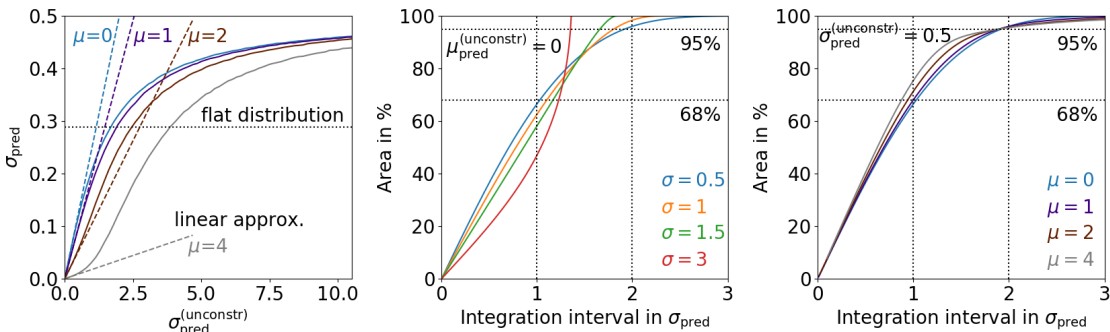

Figure 2: Left: predictive standard deviation of the after sigmoid transformation, the linear approximation is defined in Eq.(15). Center and right: area under the probability distribution in the interval $[\mu - x, \mu + x]$ for different widths and means.

a simplified detector simulation we use DELPHES [54] with the default ATLAS detector card. The fat jet is defined through the anti-$k_T$ algorithm [55] in FASTJET [56, 57] with $R = 0.8$, fulfills

$$p_{T,j} = 550 \cdots 650 \text{ GeV} \qquad \text{and} \qquad |\eta_j| < 2. \tag{12}$$

The top jets are truth-matched to a $b$-quark and two light quarks within $\Delta R = 0.8$. The images include the improved pre-processing taken from Ref. [16].

Whenever we use a neural network on classification tasks with probabilities as outputs we need to map the unbounded space of network outputs to the closed interval $[0, 1]$, for instance through a sigmoid function

$$\text{sigmoid}(x) = \frac{e^x}{1 + e^x} = \frac{1}{1 + e^{-x}} \qquad \Leftrightarrow \qquad \text{sigmoid}^{-1}(x) = \log \frac{x}{1 - x}. \tag{13}$$

Because the sigmoid is non-linear, it would not maintain a Gaussian shape of an output distribution. In Fig. 1 we show how increasingly larger widths and shifts in the mean lead to non-Gaussian features in the probability distributions on the interval $[0, 1]$ for this toy network. To compute the mean including a sigmoid transformation of a Gaussian probability distribution $G_{\mu,\sigma}(\omega)$ we solve the integral

$$\mu_{\text{pred}} = \int_{-\infty}^{\infty} d\omega \; \text{sigmoid}(\omega) \, G_{\mu,\sigma}(\omega) = \int_0^1 dx \; \frac{x}{x(1 - x)} \, G_{\mu,\sigma}\left(\log \frac{x}{1 - x}\right). \tag{14}$$

This probability distribution is known as the logit-normal distribution. For small widths of the unconstrained distributions before sigmoid, $\sigma_{\text{pred}}^{(\text{unconstr})}$, we can approximately calculate the predictive standard deviation after the sigmoid transformation as

$$\sigma_{\text{pred}} \approx \mu_{\text{pred}}\left(1 - \mu_{\text{pred}}\right) \sigma_{\text{pred}}^{(\text{unconstr})} \qquad \text{with} \quad \mu_{\text{pred}} \in [0, 1]. \tag{15}$$

In the left panel of Fig. 2 we show the correlation between the standard deviations before and after sigmoid for different means, including the linearized approximation shown above. For large values of the pre-sigmoid standard deviation the behavior of the probabilistic outcome deviates significantly from the linearized form. This is simply an effect of the interval $[0, 1]$, where we need to keep in mind that the standard deviation of a flat distribution is $1/(2\sqrt{3})$ and the largest possible standard deviation comes from a bi-polar distribution with $\sigma = 1/2$. This is exactly the plateau value we observe for $\sigma_{\text{pred}}$ for large pre-sigmoid values of $\sigma_{\text{pred}}^{(\text{unconstr})}$.

One of the questions that arises once we attempt a frequentist interpretation of the probability distributions from a BNN is the relation between the deviation from the mean in terms of standard deviations and in terms of the area under the probability distribution. We show this correlation in the center and right panels of Fig. 2. The curves indicate that below the plateau value of $\sigma_{\text{pred}} = 1/2$ the one-sigma and two-sigma limits scale reasonably well with 68% and 95% of the full integral.

Altogether, we still need to remember that the mapping from the network output over the space of real numbers to the interval $[0, 1]$ leads to non-Gaussian probability distributions. This is similar to the case of small event counts, where a Poisson distribution deviates from the symmetric Gaussian because it avoids negative event counts. Even with such non-Gaussian output the distributions before the sigmoid transformation are Gaussian, which means that they are fully described by the predictive mean and standard deviation. The network output after sigmoid is also fully described by two parameters, in our case the predictive mean and the predictive standard deviation.

## 2.3 Prior (in)dependence

As for any Bayesian setup, it is crucial that we confirm that the output of our BNN top tagger is not dominated by the prior. For the network training, the prior enters as the function $p(\omega)$ in the loss function. The trained Gaussian $q_{\mu,\sigma}$ is then constructed to fulfill the different conditions entering the loss function, Eq.(7). The default prior for our toy model is is a normal distribution with $\mu_{\text{prior}} = 0$ and $\sigma_{\text{prior}} = 1$. To test prior independence we stick to normal distributions with $\mu_{\text{prior}} = 0$, but vary the width of the prior distribution over an extremely wide range,

$$\sigma_{\text{prior}} = 10^{-2} \cdots 1000. \tag{16}$$

The setup of our classification network is described in Sec. 2.2. The performance of the different networks is measured in term of the area under the ROC curve (AUC). We find

| $\sigma_{\text{prior}}$ | $10^{-2}$ | $10^{-1}$ | 1 | 10 | 100 | 1000 |
|---|---|---|---|---|---|---|
| AUC | 0.5 | 0.9561 | 0.9658 | 0.9668 | 0.9669 | 0.9670 |
| error | — | ±0.0002 | ±0.0002 | ±0.0002 | ±0.0002 | ±0.0002 |

.

Indeed, a too narrow prior distribution does not allow the network to efficiently grasp the features of the training data. As a result, the network with $\sigma_{\text{prior}} = 10^{-2}$ does not perform at all. The situation improves towards $\sigma_{\text{prior}} = 1$ and reaches a plateau above this value. To judge the significance of the change we estimate the spread of the AUC values by training our default models five times and give the standard deviation of the corresponding AUC values. In terms of the AUC value we find a very slight systematic increase for larger prior widths, but at the price of vastly increased training time. For our toy model we have checked that also a jet-by-jet comparison of the different priors confirms the approximate prior independence of the predictive mean and the predictive standard deviation given by our default setup.

## 3 Useful features

From the construction of BNNs it is obvious that modified versions of top taggers will have some unique and attractive features for applications in LHC physics. Still based on our toy tagger we will illustrate three such features: the proper treatment of statistical uncertainties from finite training samples, an established way to calibrate the network output relative to the

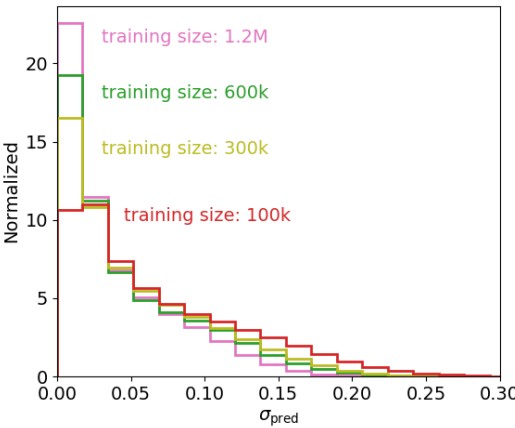

Figure 3: Normalized predictive standard deviation of signal jets for different sizes of the training sample.

accuracy on a test sample, and a proper framework of some ad-hoc features in deterministic networks.

## 3.1 Statistical uncertainty from training

To see how the the BNN setup with the predictive standard deviation works we can look at a simple source of statistical uncertainties, namely a limited number of training jets. Again, we use the fast and simple toy network to describe the main features. In Fig. 3 we show the normalized distribution of the predictive standard deviations. Running the network on the test sample of 200k top jets we histogram all predictive standard deviations for the full range of predictive means. Indeed, the distribution becomes more and more peaked for an increased training sample.

In the upper left panel of Fig. 4 we show the correlation between the predictive mean and the predictive standard deviation as the two outputs of the BNN. To construct a single correlation curve we evaluate the network on 10k jets, half of them signal and half of them background. We show the mean values of the 10k jets in slices of $\mu_{\mathrm{pred}}$, after confirming that their distributions have the expected Gaussian-like shape. The leading feature is an inverse parabola shape, which is induced by the sigmoid transform, Eq.(15). It simply reflects the fact that a network output in the interval $[0, 1]$ forces the error bars close to the ends to be comparably small. Another, physical source of the same effect is that probability outputs around 0.1 or 0.9 correspond to clear cases of signal and background jets, where we can expect the predictive standard deviation, or the error on the predictive mean, to be small. The error bars shown in the upper panels of Fig. 4 show the uncertainty on the predictive uncertainty. We derive them from five independent trainings and testings, including statistically independent samples.

In the upper right panel we illustrate the improvement of the network output with an increasing amount of training data by showing the predictive standard deviation for $\mu_{\mathrm{pred}} = 0.45 \cdots 0.55$ as a function of the size of the training sample. The estimated uncertainty on the tagger output decreases monotonically from 16% to 12% when we increase the training sample from 100k to 1.2M jets. This improvement is significant compared to the error bars, which correspond to different training and testing samples.

Finally, the spread of these 10k signal and background jets is illustrated in the four lower panels, with a matching color code. We immediately see that the spread is strongly reduced for larger training samples.

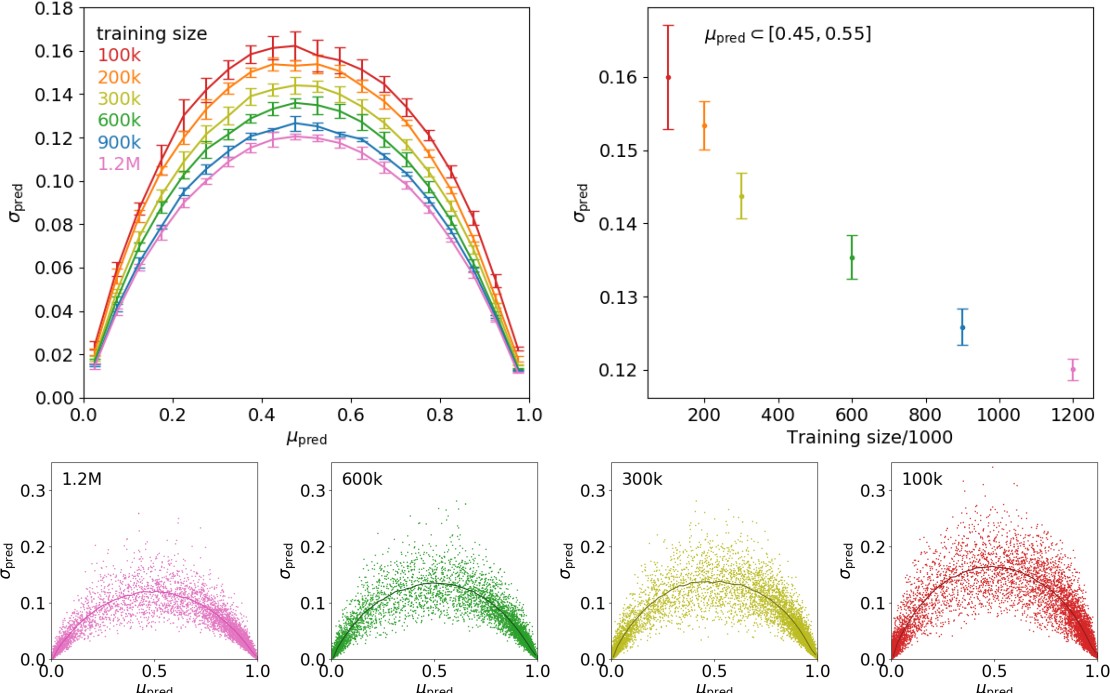

Figure 4: Correlation between predictive mean and standard deviation. The error bars in the upper left panel correspond to five independent trainings and indicate the uncertainty on the uncertainty given by the BNN. The right panel shows the predictive standard deviation for $\mu_{\text{pred}} = 0.45 \cdots 0.55$ as a function of the size of the training sample with the same error bars from different trainings. The lower panels instead show the statistical spread for 10k jets, signal and background combined.

## 3.2 In-situ calibration of weight distribution

Before we attempt to compare the output of the BNN to a frequentist distribution of many deterministic neural networks we can apply a cross check within the Bayesian framework itself and construct a hybrid version of the BNN. This will lead us to another attractive feature of such networks, their explicit calibration based on training data.

The standard BNN constructs its output distribution by sampling the individual weights of each layer. They are initialized as a random set of Gaussians with different means and standard deviations, which are then learned during training. This means that the BNN not only learns a set of weights, but a distribution of weights from the first layer to the network output.

To generate a distribution we can also train the BNN a large number of times and only use the maximum values or means of the (Gaussian) weight distributions. This should encode the same information as the BNN, just much less efficiently. One of the problems with this so-called maximum-a-posteriori (MAP) approach is that it does not include a Bayesian integral over weight space and instead works like a frequentist profile likelihood. This profiling does not maintain the normalization of the probability distribution and therefore makes it impossible to naively compare for example ranges of the network output of the kind $60 \cdots 70\%$ vs $80 \cdots 90\%$ top jet probability. To re-gain this interpretation and to be able to compare the full BNN with this MAP approximation we can re-calibrate the tagger [58]. This re-calibration ensures that the probabilistic output of the network and the measured accuracy on a test sample are aligned, *i.e.* an 80% top jet probability network correctly identifies 80% of top jets in a given sample. Obviously, it is a tool which can be applied much more generally than for our toy comparison.

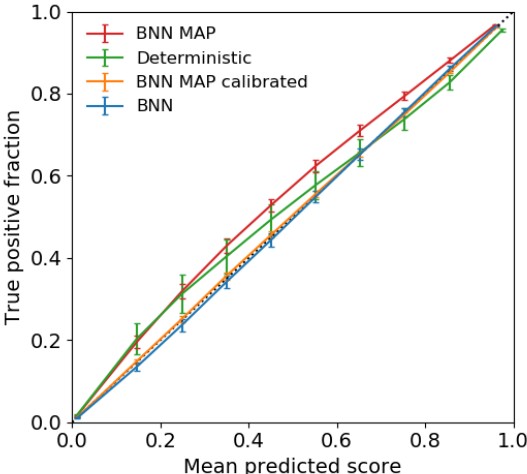

Figure 5: Reliability diagrams of different BNN approaches, as well as the set of deterministic taggers discussed in Sec. 3.3. The bin-wise true positive rate is plotted against the mean prediction of the jets in one bin. The error bars indicate the spread for 50 independent training samples.

The effect of such a re-calibration can be shown in reliability diagrams like in Fig. 5. They are constructed by discretizing the tagger output, *i.e.* the signal or background probability, into several bins of equal length and correlating this number with the true positive rate in each bin. In a well-calibrated model the resulting curve would sit on the diagonal, modulo uncertainties. Using 400k top and QCD jets integrated over ten bins we see that unlike the consistent BNN output, the MAP output is poorly calibrated. The error bars indicate the spread for 50 independent training samples. We find that it can be significantly improved through Platt scaling [59]: here we first transform the before-sigmoid output linearly, $x' = a x + b$. The free parameters $a$ and $b$ can then be determined by minimizing the usual cross entropy for fixed weights on a validation set. Indeed, the re-calibrated MAP network is well calibrated and can be used to extract a sensible tagging output. Obviously, this kind of re-calibration between the network output and the measured purity is not only useful to study the behavior of our toy BNN, but it can be used to calibrate any classification tool, for instance at the LHC.

In Fig. 6 we show the correlation between the predictive mean and predictive significance from the BNN output and using the MAP results. Each point in the scatter plot corresponds to one of 4000 jets. Indeed, the predictive BNN output scatter the same way as the weights after re-calibration. This implies that, in spite of the Bayesian setup, we can think of the predictive standard deviation as representing the distributions of the network weights in a frequentist sense.

## 3.3 Relation to deterministic networks

To investigate a frequentist interpretation of the BNN output we will compare the predictive mean and standard deviations with the corresponding distributions for a large number of independently trained and tested deterministic networks. The architecture of the deterministic model will have the same number of layers and nodes as the BNN. In the deterministic network we minimize the negative logarithm of the likelihood defined in Eq.(2) or, equivalently, the cross-entropy. Applying an additional L2-regularization allows us to drive the network to

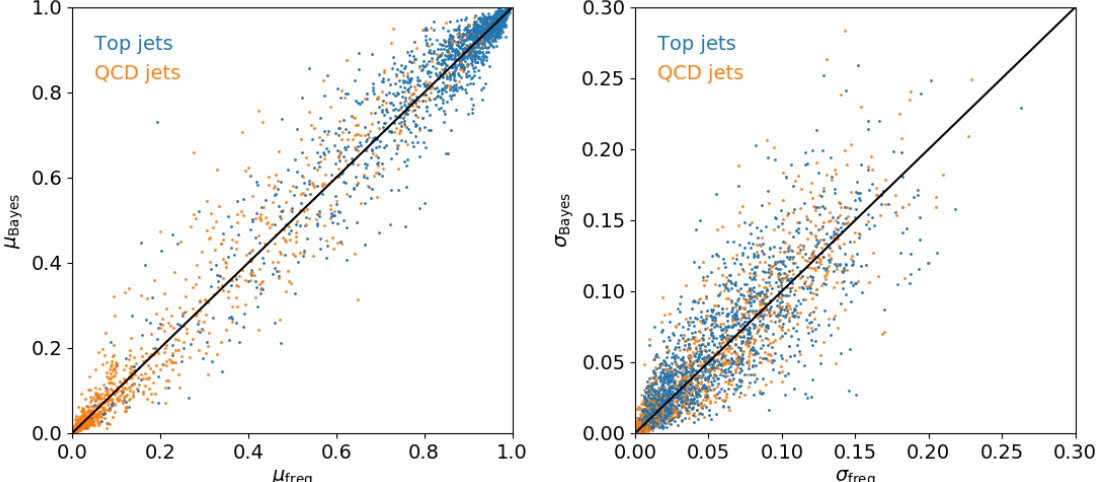

Figure 6: Correlation of the mean (left) and standard deviation (right) between the BNN's $\mu_{\text{pred}}$ and $\sigma_{\text{pred}}$ and the histogrammed weights fixed to their respective means. Each point corresponds to one of 2000 signal and 2000 background jets, as indicated by the colors.

more stable or less complex models,

$$L = -\log p(C|\omega) - \lambda|\omega|^2 \equiv \frac{\chi^2}{2} - \lambda|\omega|^2, \tag{17}$$

where we give the relation to the $\chi^2$ in the simplest Gaussian approximation. In a Bayesian approach we instead minimize Eq.(7), a combination of the negative expected log-likelihood and the KL-divergence,

$$L = -\log p(C|\omega) - \frac{\mu^2}{2\sigma_{\text{prior}}^2} + \cdots. \tag{18}$$

Relating these two approaches and identifying $\omega$ in the deterministic network with the $\mu$ in the BNN suggests that a Gaussian prior corresponds to an L2-regularization,

$$\lambda = \frac{1}{2\sigma_{\text{prior}}^2}. \tag{19}$$

At this point it is crucial to emphasize that the deterministic network has this additional option of balancing the network performance, training time, and stability with the help of the free L2-regularization parameter $\lambda$. In contrast, for the BNN $\sigma_{\text{prior}}$ is fixed by the choice of prior.

For this comparison, we require considerably larger statistics than is available in the public dataset based on Ref. [17]. We generate an new data set in the same phase-space region as in Sec. 2.2, but with 9.5M top jets and 9.5M QCD jets. We then use a deterministic toy tagger with the same architecture as the toy BNN, but without the Bayesian features. We construct 100 different networks training on statistically independent samples and histogram their classification output for a given jet. From this histogram we can extract the mean and the standard deviation in a frequentist sense. Because the Bayesian and deterministic networks have the same setup, we expect this distribution of 100 means to follow the prior-independent probability distribution from the BNN. In Fig. 7 we show the correlation between the predictive mean and predictive significance with the results from a set of deterministic taggers. Each point in the scatter plot corresponds to one of the 4000 jets. Jet by jet we see that the Bayesian and

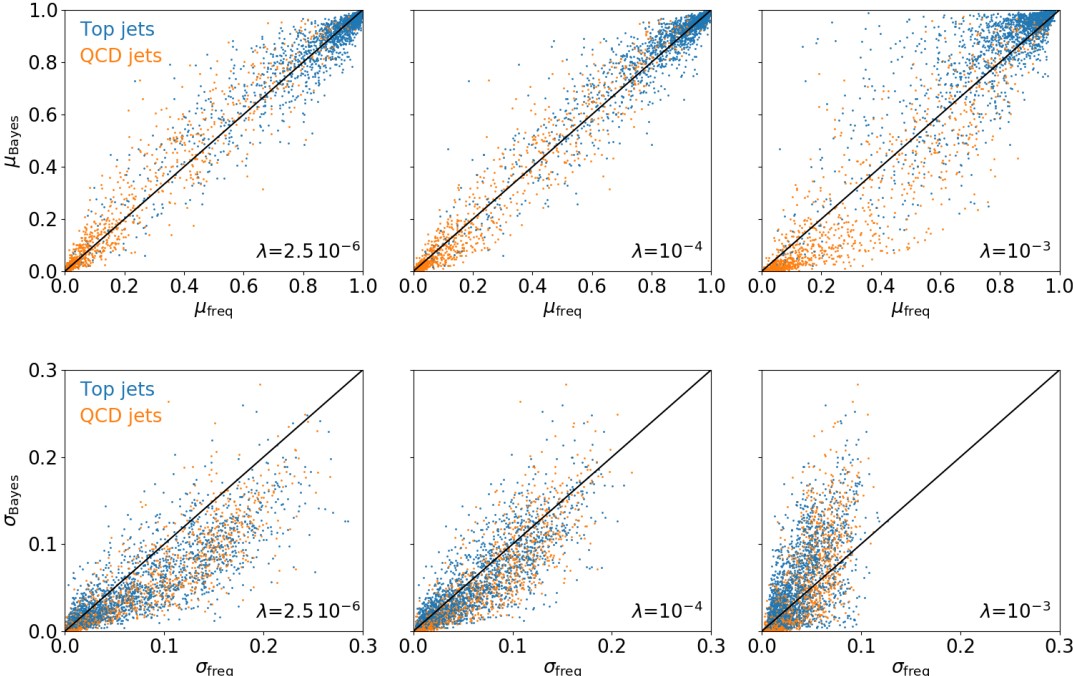

Figure 7: Correlation of the mean (upper) and standard deviation (lower) between the BNN's $\mu_{\text{pred}}$ and $\sigma_{\text{pred}}$ and histogrammed outputs for 100 deterministic networks. We show results for three different values for the L2-regularization $\lambda$. Each point corresponds to one of 2000 signal and 2000 background jets, as indicated by the colors.

frequentist mean values are clearly fully correlated, with an increased spread for relatively poorly determined values around $\mu = 0.5$. This spread is expected, and we have checked that it is within the range of the predictive standard deviation. The slight asymmetry of the taggers on the pure top and QCD sides is not an unexpected feature. The default value for the deterministic L2-regularization given in Eq.(19) in our case is

$$\lambda = 2.5 \cdot 10^{-6}, \tag{20}$$

shown in the left panel of Fig. 7. We also show how much larger values of $\lambda$ lead to a significant underestimate of the uncertainties in the frequentist approach, corresponding to a strong prior-like behavior of the L2-regularization. The apparent agreement of the two approaches not only in the mean but also in the width is best for $\lambda = 10^{-4}$, while for the default value of $\lambda$ the deterministic network give a slightly more conservative error band.

The discussion above leads us to another freedom we have in defining our deterministic tagger: while we have not used it till now, dropout reduces the number of neurons in each training epoch statistically to a given percentage and is used to avoid over-fitting [60]. If a neuron is switched off for one iteration, the other neurons will compensate for this loss, leading to a stochastic fluctuation within the network from iteration to iteration. The key aspect of dropout is that it generates statistical fluctuations in the network weights, and it can be shown that networks trained with dropout are BNNs [61]*. In the absence of a quantitative relation like Eq.(19) we have to test the dependence of our comparison between the BNN

---

*Strictly speaking, all networks using dropout are Bayesian networks, but not all Bayesian networks can be modelled based on dropout.

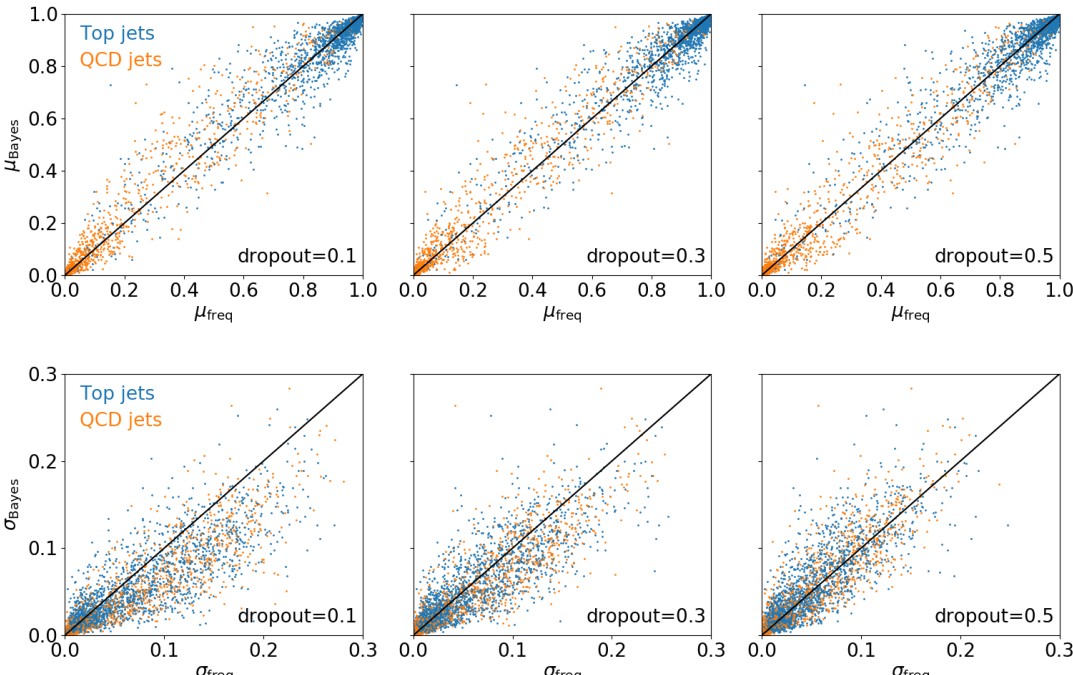

Figure 8: Correlation of the mean (upper) and standard deviation (lower) between the BNN's $\mu_{\text{pred}}$ and $\sigma_{\text{pred}}$ and histogrammed outputs for 80 deterministic networks with $\lambda = 2.5 \cdot 10^{-6}$. We show results for three values for the dropout rate. Each point corresponds to one of 2000 signal and 2000 background jets, as indicated by the colors.

and the deterministic networks on the dropout rate of the deterministic networks. For the deterministic networks dropout is used during training but not during testing. In Fig. 8 we show three different correlations for dropout rates of 0.1, 0.3, and 0.5, for the default L2-regularization. Again, we see that small dropout leads to a slightly larger frequentist estimate of the uncertainties, while large dropout has a prior-like effect of reducing the frequentist error estimate. In the center panel of we see that

$$\lambda = 2.5 \cdot 10^{-6} \qquad \text{and dropout rate } 0.3 \qquad (21)$$

lead to an excellent agreement of the jet-by-jet BNN output and the frequentist analysis of 100 deterministic networks. In practice, the difference between these two approaches is that training a large number of deterministic networks on an actual tagging setup is extremely GPU-intensive, while the BNN provides $\sigma_{\text{pred}}$ on a jet-by-jet basis automatically with the probabilistic mean $\mu_{\text{pred}}$.

Aside from the fact that we can reproduce the probability distribution of tagging output in a frequentist sense there is another, more conceptual lesson to learn from the this section: while the loss function of Eq.(7) or (18) comes from Bayes' theorem, the appearance of the KL-divergence is closely linked to known numerical improvements of standard deterministic networks, like dropout and L2-regularization. In that sense, the BNN with its jet-by-jet distribution of tagging outputs is not any more Bayesian than many of the NN-taggers which we already use.

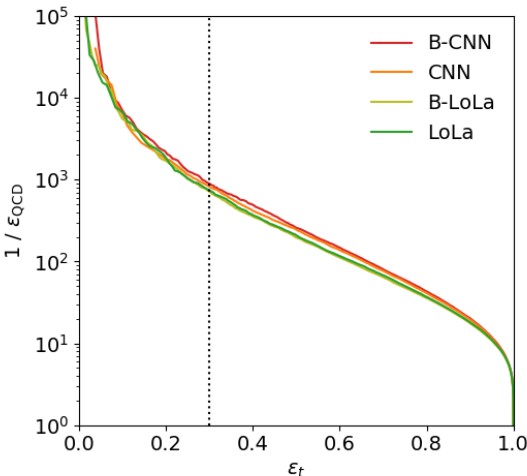

Figure 9: ROC curve for the deterministic image-based and LOLA top taggers, compared to their respective BNN implementation.

# 4 BNN top taggers

After understanding the concept behind BNNs and illustrating many of their features, we are now ready to apply them to an actual physics problem. Top tagging is an established, experimentally and theoretically well-defined task [24, 62]. Experimentally, it has the great advantage that we can train neural networks on mixed top pair events, where we first identify the leptonically decaying top quarks and then run the tagger on the hadronic recoil. Theoretically, the main features of top decays are safely perturbative [17], so unlike, for instance, in the case of quark-gluon separation [10] we do not expect detector effects and pile-up to have decisive impact on the tagging performance. On the other hand, detector effects and (related) systematic effects are going to be key factors in any application of machine learning techniques to subjet physics [25], especially if we eventually need to go beyond perfectly labelled actual jets to MC-enhanced training samples.

Table 1: Performance of the different tagging architectures and their BNN versions on our standard top sample.

|  |  | AUC | $1/\epsilon_{\text{QCD}}$ for $\epsilon_t = 30\%$ |
|---|---|---|---|
| CNN |  | 0.982 | 820 |
| B-CNN |  | 0.982 | 900 |
| LoLa | $N_{\text{const}} = 40$ | 0.979 | 630 |
| B-Lola |  | 0.979 | 600 |
| LoLa | $N_{\text{const}} = 100$ | 0.981 | 740 |
| B-Lola |  | 0.980 | 710 |
| LoLa | $N_{\text{const}} = 200$ | 0.980 | 600 |
| B-Lola |  | 0.980 | 710 |

## 4.1 Performance

To test how BNNs with different architectures react to detector effects or systematic uncertainties, we implement BNN versions of an image-based DEEPTOP tagger [15, 16] and the 4-vector-based DEEPTOPLOLA tagger [17]. While those two specific implementations do not show the leading performance in top tagging [24], they represent their respective architectures with a good compromise between performance and run time.

Our test sample is the same top and QCD data set with 200k jets [17] as described in Sec. 2.2. Again, the jets fulfill

$$p_{T,j} = 550 \cdots 650 \, \text{GeV} \qquad \text{and} \qquad |\eta_j| < 2. \tag{22}$$

The top jets are truth-matched, and the images include the improved pre-processing taken from Ref. [16]. The constituents for the LOLA tagger are extracted through the Delphes energy-flow algorithm, and the 4-momenta of the leading 200 constituents are stored. For jets with less than 200 constituents we simply add zero-vectors.

In Tab. 1 we show the performance of the different deterministic taggers [15–17], as well as of their BNN counterparts. For this application we vary the number of ordered particle flow constituents considered by the LOLA tagger between 40 and 200. For example in Fig. 2 in Ref. [17] we see that the number of constituents in top jets is around 60, with a sizeable tail towards significantly larger numbers. On the other hand, we also know that many of them correspond to soft activity, which according to QCD factorization is universal and essentially adds noise. This theoretical bias is confirmed by the performances shown in Tab 1, where the LOLA tagging performance indicates no significant improvement once we increase $N_{\text{const}}$ beyond 40.

In Fig. 9 we see the same behavior when we consider the entire ROC curve for top tagging in the presence of a QCD background. Within uncertainties related to different trainings both taggers and their BNN counterparts show essentially the same performance. Note that this statement only holds true in the absence of pile-up and if we ignore statistical and systematic uncertainties. Statistical uncertainties include, for instance, the effect of the size of the training sample, as discussed in Sec. 3.1 and Fig. 4. We do not repeat this exercise for the actual taggers and instead focus on detector and systematic effects. As a starting point, we discuss how systematics and detector effects, not accounted for in the training, affect the predictive mean and standard deviation given by the BNN. Finally, we show how the BNN works if we use for example a known systematic uncertainty to modify or augment the training data.

## 4.2 Systematic uncertainty from energy scale

As part of our program of including uncertainties through BNNs, we can investigate the jet energy uncertainty as an actual systematic uncertainty [63,64]. This means that the jet energy scale has to be calibrated with standard candle processes, and this calibration comes with an uncertainty from the underlying measurement and from the extrapolation to a given event topology. We use the data set described in Section 2.2 and focus on a cluster re-scaling of the constituents of the fat jet. It involves re-clustering the constituents into anti-$k_T$ subjets with $R = 0.4$ and in the simplest, toy case re-scaling the energy of the leading subjet cluster. While the actual jet energy scale uncertainty at the LHC is in the range of 1% $\cdots$ 3%, we inflate the shift in this section to illustrate its features for our networks limited by GPU time and number of training jets. This toy model for smearing is not meant to be experimentally realistic, but it is chosen to identify the non-trivial features which occur in the presence of unknown systematics.

In a first attempt we test what happens if we train on data which does not account for a given systematics, but we test the trained network in the presence of a systematic shift, in

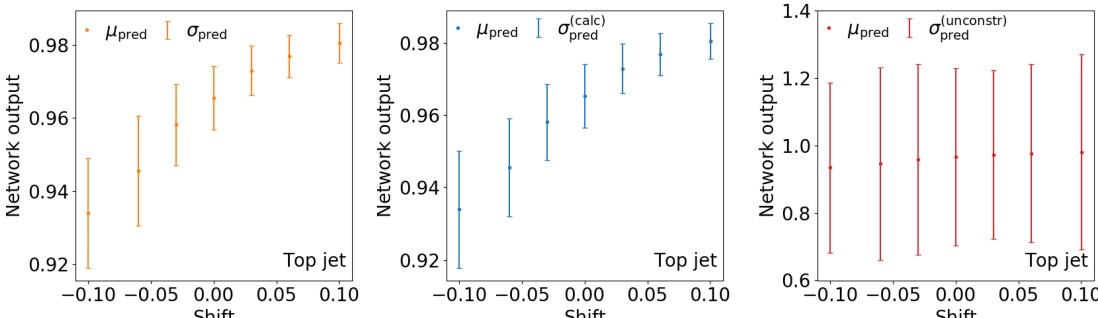

Figure 10: Effect of a shifted energy scale for the hardest constituent on the mean and the standard deviation indicated as an error bar. We show the probability output (left), supplemented with the effect on the predictive uncertainty through the mean (center), and before the sigmoid transformation (right), all for the BNN LoLa tagger.

this case shifting the energy scale of the leading constituent in top and QCD jets by up to 10%. For this study we use a BNN version of the LoLa tagger, because CNN taggers usually normalize the pixel entries relative to the total momentum. In contrast, for the LoLa tagger we can reduce the error bars due to statistics of the training sample to the level where we can see systematics-induced shift in the discrimination power and the assigned error bars. In Fig. 10 we see the effect of such an energy rescaling on the network output for one given jet. For one sign of the energy shift we see the expected behavior, namely a loss in tagging performance combined with an increased assigned error bar. For the other sign of the shift we see a pattern which naturally arises whenever the systematics is not fully de-correlated from the features the network uses to separate the two hypotheses. In our case, we know for instance that the network will identify jets with a more democratic energy distribution of the constituents with QCD. This is why a positive shift of the leading constituent leads the network to more confidently identifying the jet as a top jets, including a small error bar. This behavior is similar to adversarial examples [65], single-pixel modifications meant to trick image recognition tools into wrong classification obvious to human vision.

From Sec. 2.2 we know that the network output after the sigmoid transformation strongly correlates the mean and standard deviation. To understand the effect shown in the left panel of Fig. 10 we therefore use Eq.(15) to determine the shift in the predictive standard deviation based on this correlation with the shifted mean. In the center panel we indeed see that this correlation dominates the change in the predictive standard deviation for this one jet. Alternatively, we can instead show the standard deviation before the last sigmoid layer as a function of the energy shift, confirming this picture in the right panel.

In a second step we can now look for changes in the predictive standard deviation which are independent of the shift in the mean. For this purpose, we train our BNN without energy smearing and test it on jets with a smeared jet energy. For further de-de-correlation we add a 4-vector normalization to the LoLa TAGGER, such that the energy sum of all constituents per fat jet is one. This is a standard step in the image-based taggers. Unlike before, we then re-cluster the constituents into anti-$k_T$ subjets and re-scale the energy of each subjet (or cluster) by a random number drawn from a Gaussian distribution with mean zero and a given standard deviation. After this shift the predictive uncertainty given by the network before sigmoid becomes $\sigma_{\text{smear}}^{(\text{unconstr})}$, and we show the distribution for the normalized shift

$$\frac{\sigma_{\text{smear}}^{(\text{unconstr})} - \sigma^{(\text{unconstr})}}{(\sigma_{\text{smear}}^{(\text{unconstr})} + \sigma^{(\text{unconstr})})/2}.$$

(23)

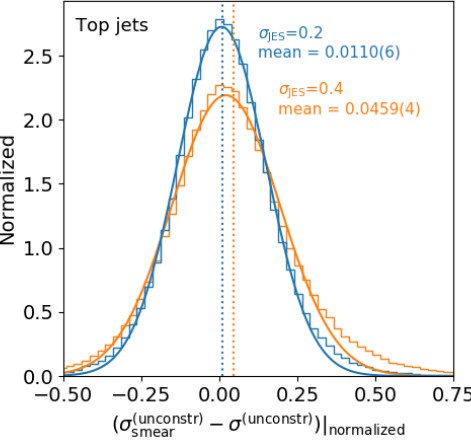
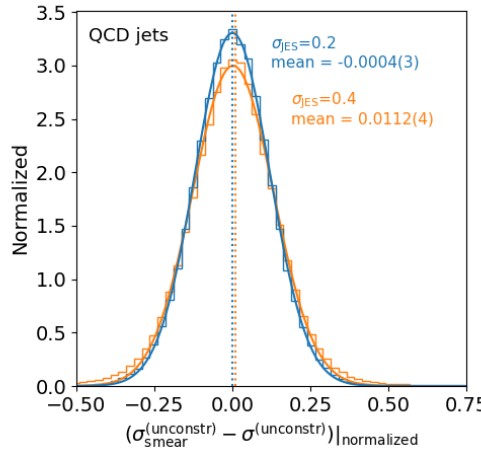

Figure 11: Effect of a shifted jet energy scale on the predictive standard deviation output before sigmoid for the BNN LoLa tagger. We show top jets (left) and QCD jets (right) separately.

In Fig. 11 we show this effect separately for a sample of top and QCD jets. For the top jets we see how for a jet energy smearing of 20% the predictive standard deviation increases, but very slightly. The fit to a symmetric Gaussian indicates how the tail towards larger widths becomes bigger than the tail towards smaller width. The mean of the distribution confirms a small, but significant shift. As a test, we also show the numbers for a cluster-wise shift of 40% in the jet energy scale, leading to a consistent but larger effect. In the right panel we show the same for QCD jets. Here, at least a 20% jet energy re-scaling does not produce a visible effect. The reason is that QCD jets have a comparably democratic distribution of subjet energies, so a random smearing has no effect of the QCD-ness in the eyes of the network unless we apply a 40% smearing.

Once we understand the jet energy scale systematics, we naturally want to include them in the training. This means we will take actual training data and manipulate it through a smearing procedure which mimics the jet energy scale uncertainty. We analyze the impact of such a data augmentation for our cluster model of the jet energy scale systematics by training the network on data after applying a Gaussian smearing with a width of up to 10%. Unlike before, we now smear all subjet clusters after re-clustering, sampling from a Gaussian. In the left panel of Fig. 12 we first see how training with a 10% smearing in the jet energy scale affects the tagging output on a typical top jet. Indeed, training on augmented data stabilizes the tagging output compared to the situation shown in Fig. 10. Specifically, we see that training on 10% smearing pushes the drop in tagging performance to beyond that value, while it does not affect the otherwise stable performance. This is not a unique feature of the BNN, but also present for the deterministic tagger. In the right panel of Fig. 12 we see that training on strongly smeared data will, evaluated on un-smeared samples, lead to a decrease in performance. This is not surprising, because smearing washes out features. However, tested on smeared data augmented training data leads to a visible gain in stability. While given our numerical limitation we only show this for unrealistic shifts in the jet energy scale, the feature in itself exists, and according to the left panel of Fig. 12 it is significant.

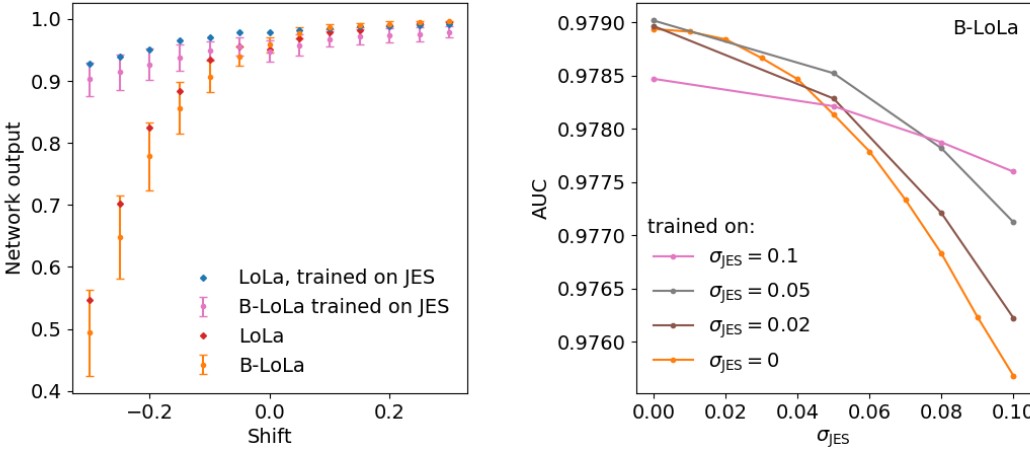

Figure 12: Change of the LoLa tagging performance as a function of a jet energy smearing after training on unsmeared or augmented data. Left: network output for a single top jet indicating the predictive standard deviation as an error bar. The training sample with JES smearing uses $\sigma_{\text{JES}} = 10\%$. Right: AUC for a top and QCD sample for different amounts of JES smearing in the training sample.

## 4.3 Stability tests for pile-up

Finally, we can test how BNNs deal with systematic effects in the data, which are not accounted for as an uncertainty. For instance pile-up is known to lead to problems for subjet physics tools. Strictly speaking, pile-up is not a systematic uncertainty, because we can measure it statistically and even attempt to remove it event by event. It arises from a high multiplicity of interactions per bunch crossing. To remove it, some standard tools such as PUPPI [66] and SOFTKILLER [67] are typically employed, and more recently there has been successful applications of machine learning methods to this problem [68,69]. In this section we test how stable deterministic and BNN approaches are to the amount of pile-up in an event with jets.

To simulate pile-up, we generate min-bias events again with PYTHIA8 [53], including DELPHES [54] and with the same settings as in Ref. [15]. In total, we generate 1M min-bias events with the hardest 400 constituents of each event. We then add a variable number of up to 200 min-bias events on top of the signal and background jet events. For this combination we re-cluster the fat jets and select the hardest jet per event. The new jets are pre-processed the same way as the jets without pile-up described in Sec. 4.1. For the stability test we train the networks on a sample without pile-up and test it on samples with different amount of pile-up. We note that this comparison ignores the fact that we can actually simulate pile-up and that experimental training data tends to include pile-up at some level, so all we test is the stability of the taggers.

In Fig. 13 we show the background rejection at fixed signal efficiency $\epsilon_t = 30\%$ as a function of the number of pile-up events added. While for the CNN and its BNN version we see no significant difference in performance, the LoLa tagger shows some interesting features. The reason for this different behavior is the fact that in the latter the constituents are not just added noise, but given in an ordered manner. As long as we only use $N_{\text{const}} = 40$ constituents the LoLa tagger simply ignores the relatively soft contributions from pile-up. In that sense it is by construction insensitive to universal soft activity, independent of its source. We have confirmed this pattern for our tagger.

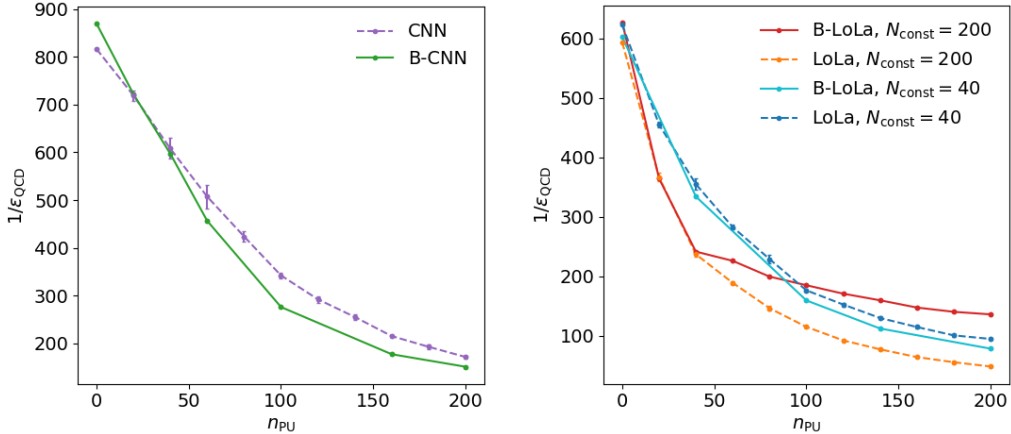

Figure 13: Impact on pile-up for the image-based tagger (left) and the LOLA tagger with two numbers of constituents (right).

The picture changes when we start to include up to $N_{\mathrm{const}} = 200$ constituents. In Fig. 13 we see how the deterministic tagger is now sensitive to noise effects, and how an increased amount of noise cuts into the tagging performance. However, the behavior of the BNN version of the LOLA tagger is different; here, the BNN setup allows the tagger to utilize the additional information, even though there exists only soft QCD radiation related to a single bunch crossing in the training, and turn it into a performance improvement. In Fig. 14 we see, however, that this improvement of the performance comes at a price in the uncertainty. For an increased number of pile-up events the predictive standard deviation increases from 12% to 22%. In the lower four panels we see that this improvement happens in a perfectly stable system, maintaining the expected parabolic correlation between the mean and standard deviation outputs of the tagger.

Finally, we can apply the same test to the BNN version of the image-based tagger. According to Fig. 13 the performance of the CNN taggers is relatively stable with respect to pile-up. In Fig. 15 we show the performance of the Bayesian CNN tagger in more detail and immediately see that for pile-up values of 60-100 the network becomes unstable and violates the parabolic shape predicted for the probabilistic output in Eq.(15). This parabolic correlation, however, is a fundamental effect of the network output on the closed interval $[0, 1]$. The deviations start affecting the more poorly classified jets and is not immediately visible for example from the uncorrelated output of the predictive mean and the predictive standard deviation. This means that the careful analysis of the predictive standard deviation allows us to gain addition insight on the stability of the network, which we do not gain from a performance study of the deterministic network.

# 5 Outlook

Machine learning applied to low-level detector observables has the potential to transform many aspects of LHC analyses, for instance subjet analyses and top tagging. Two open questions for instance of classification networks, independent of their architecture and setup, are how we can include a proper error treatment and how we can understand the network output. Bayesian neural networks offer solutions to both of these problem, also going beyond what established tagging approaches can offer. Their list of advantages and opportunities for LHC applications is remarkable: (i) they can classify jets or events including error bars; (ii) they

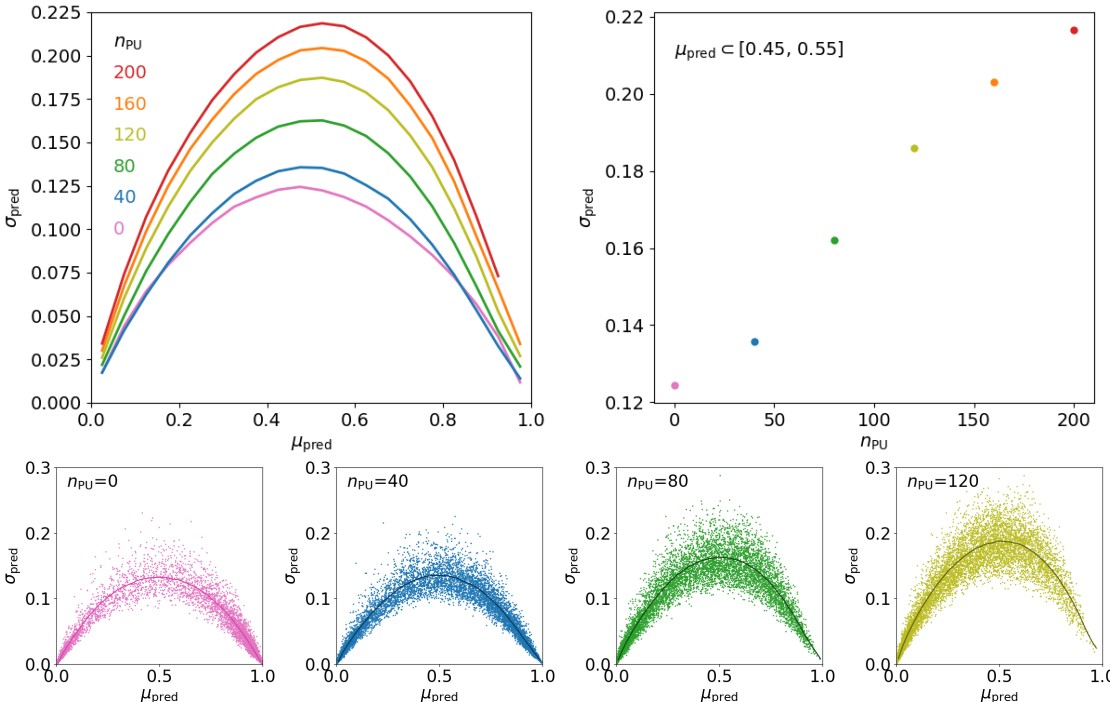

Figure 14: Correlation between predictive mean and standard deviation for the BNN
LoLa network with pile-up included. The right panel shows the predictive standard
deviation for $\mu_{\text{pred}} = 0.45 \cdots 0.55$ as a function of the number of added pile-up
events. The lower panels instead show the statistical spread for 10k jets, signal and
background combined.

can can be interpreted in a frequentist sense, the prior has no visible impact; (iii) they help
us understand regularization and dropout in deterministic networks; (iv) they are usually
well-calibrated, even though re-calibration can always streamline LHC applications. We have
illustrated these features and their application to LHC physics using a toy top tagger working
on jet images.

The standard way classifiers in HEP analyses are currently employed is to determine a
working point and then assess the uncertainty on its signal efficiency and background rejection.
A classifier with per-jet or per-event uncertainties automatically provides these as well but with
additional information provided by the uncertainty which could be potentially included in a
statistical analysis. We have shown that standard deep-learning tagging frameworks, like an
image-based CNN tagger or a 4-vector-based LoLa tagger, can be easily extended to a BNN
version. Specifically, we have shown how the Bayesian versions

- track the statistical uncertainty from a limited training sample, Fig. 4;

- track mean-correlated systematics from the jet energy scale, Fig. 10;

- track systematics orthogonal to the correlation with the mean, Fig. 11;

- provide a handle on the stability with respect to pile-up, Fig. 13;

- have the same performance as deterministic networks, Fig. 9.

We note that all the above advantages of the BNN extensions are available at essentially no
cost, but they provide a wealth of additional information and insight into the behavior of deep
networks. Especially for notorious users in particle physics these additional handles controlling
and understanding the network output should be very attractive. They might also be useful in
determining the best-suited classification network architectures for ATLAS and CMS.

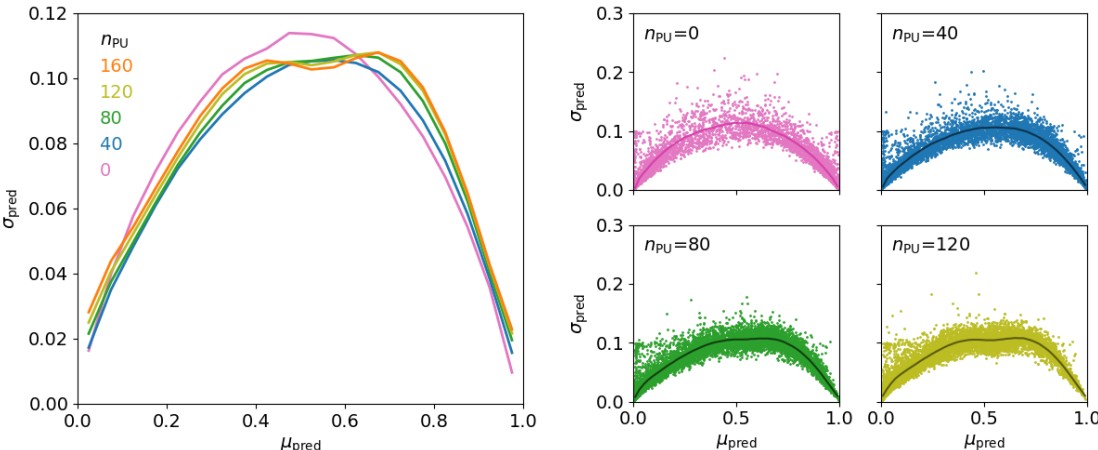

Figure 15: Stability of the image-based BNN in the presence of pile-up, in analogy to Fig. 14.

# Acknowledgments

We would like to thank Fred Hamprecht and Ullrich Köthe for brokering this collaboration and for their support. We could also like to thank Ben Nachman for some very useful discussions on deep learning and uncertainties, and Kyle Cranmer for pointing us to some early papers. The authors acknowledge support by the state of Baden-Württemberg through bwHPC and the German Research Foundation (DFG) through grant no INST 39/963-1 FUGG. JT would like to thank BMBF for funding.

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
