# Peer review of "Deep-Learning Jets with Uncertainties and More"

_SciPost Physics, doi:SciPost Phys. 8, 006 (2020)_

## Round 1 · Referee Report · Anonymous · 2019-6-3

Strengths
1- Applied an interesting idea from ML (BNN) to HEP and provided explanation and intuition for what and how the new technique is able to accomplish.
2- Used an important HEP concept to illustrate the new application (top tagging).
3- Explored the dependence of the application on topics relevant to HEP (limited training samples, systematic uncertainties, pileup).
Weaknesses
1- The notation is at times a bit confusing and potentially misleading. For details, see the Report.
2- I left this not knowing exactly how these BNNs will be used in practice. For the central values, it is nice that this can be mapped onto a particular regularization on the weights. This may help with robustness and so the application is clear. However, how will the error bar be used? This is not obvious - see more details in the Report.
Report
- Eq. 3 has a mistake: $p(c^*|\omega)$ on the righthand side should be $p(c^*|\omega,C)$. Related: I am not sure what $p(\cdot | \omega)$ means - the parameters $\omega$ are meaningless without also specifying a model $C$. Please fix/explain. Since this features prominently in Fig. 7+, it is important to clearly state what is meant.
- "as long as we can show that the final result does not depend on its form." - please decide if you are a Bayesian or a Frequentist; if you are a Bayesian, then there is no reason why the final result should be independent of the prior (the prior is supposed to encode actual a priori information).
- Fig. 1a: I thought that the network used ReLU - how can the output then be negative? Perhaps this has to do with the unexplained DenseFlipout layer?
- Fig. 6+: Why is the distribution different for top and QCD jets? I would have thought that the weight distributions are fixed after the training and do not depend on the examples (maybe this is a place where removing the D from the notation is actually hurting more than helping).
- "Aside from the fact that we can reproduce..." this is one of the most important and interesting paragraphs in the whole paper - perhaps it is worth mentioning this in the abstract / intro? (it is briefly mentioned in the conclusions).
- Fig. 13b: I did not follow why there is a kink for the B-LoLa for 40 constituents.
- Can you please explain how the error bar will be used in practice? Of course, the tagger can be calibrated using data and the uncertainty on the calibration will be the uncertainty on the NN - not the uncertainty from the BNN itself. It is interesting that you now have some measure of "confidence" in the NN decision, but it is not related to the tagger uncertainty.
- Perhaps it would be useful to make a connection to other methods proposed to assign observable error bars such as QJets (1201.1914) and Fuzzy Jets (1509.02216)?
Requested changes
1- Please see all of my questions and comments in the Report. They are mostly minor.

---

## Round 1 · Referee Report · Anonymous · 2019-6-14

Strengths
1. Very topical. Interest in the topic has moved on from naive classification performance to stability, generalization, uncertainties and this paper makes definite advances in some of these directions.
2. Pedagogical structure with lots of intuition-building.
3. The comparison with frequentist uncertainties is reassuring.
4. Clear and attractive figures.
Weaknesses
See report.
Report
This is a good paper, definitely a valuable addition to the quickly growing literature on Machine Learning for LHC physics, and I recommend publication subject to minor revisions.
1. You seem to have missed the 2007 Masters Thesis of Saucedo, "Bayesian Neural Networks for Classification" which appears to be the first published study of BNNs for classification in particle physics (for the Tevatron in this case) https://fsu.digital.flvc.org/islandora/object/fsu%3A180312. This paper should be cited, and you should clarify how you differ from/go beyond this earlier work. I also recommend revisiting the sentence "In this paper we will, for the first time, introduce Bayesian networks to a standard classification task in particle physics".
2. It would be helpful if you could clarify a little more the utility of event-level or jet-level classification error bars. While it certainly seems like it should be a useful quantity, physics searches have been successful so far in assessing systematic uncertainties on selection efficiencies etc using classifiers that do not intrinsically output event- or jet-level uncertainties. It is great seeing how the error bars relate with training statistics and smearing in intuitive ways and can therefore be a helpful diagnostic but I wonder whether it can be used to actually improve the precision/accuracy of an SM measurement or the reach of a BSM search.
3. I was a little confused in section 3.3 on the dropout discussion. Dropout is typically applied during training but not during testing. But in the case of ref [31] it is also applied during testing so that the score is not deterministic. I think the authors of this paper applied dropout during training but not testing but I'm not 100% sure. The discussion and results are the same either way but maybe this can be made completely explicit. I think not applying dropout during testing is like BNN MAP discussed earlier in the paper.
Requested changes
See Report.

---

## Round 2 · Referee Report · Anonymous · 2019-11-20

Report

The revision has addressed all of my minor concerns listed in my previous report, and I recommend publication at this point.

---

## Round 2 · Referee Report · Anonymous · 2019-11-27

Report

Thank you for your responses and for v2! I have just two residual comments, hopefully it should be quick to address them:

Comment on v1: Fig. 13b: I did not follow why there is a kink for the B-LoLa for 40 constituents.

Your answer: We looked in more detail into this. The sharp kink is statistics. Training several times and including more points around 40 constituents leads a more smooth curve. We believe that the improvement starts at around 40 constituents, because this is a point where the top and QCD jets of our samples start to differ significantly from the once without pile up. We haven't included this explanation to the paper.

My response: This seems like a reasonable answer to me - why not add a note about it in the paper?

Comment: Can you please explain how the error bar will be used in practice? Of course, the tagger can be calibrated using data and the uncertainty on the calibration will be the uncertainty on the NN - not the uncertainty from the BNN itself. It is interesting that you now have some measure of "confidence" in the NN decision, but it is not related to the tagger uncertainty.

Your answer: We included a sentence to the conclusion: "The standard way classifiers in HEP analyses are currently employed is to determine a working point and then assess the uncertainty on its signal efficiency and background rejection. A classifier with per-jet or per-event uncertainties automatically provides these as well but with additional information provided by the uncertainty which could be
potentially included in a statistical analysis". This doesn't answer the question and rather states that we currently don't know. However, this doesn't mean there is no way to include the additional information we get from the jet level uncertainty to an actual analysis. But this is probably beyond the scope of this paper.

My response: I think it is fine that you don't know how it will be used, but I don't agree with what you wrote. The uncertainty from the calibration on a tagger comes from the precision on the data/MC scale factors. What you have computed is a statistical "uncertainty" related to the training. These are not the same thing. Please reword.

Requested changes

Please see the report.

---

## Editorial Decision

published